# Practical Determination of the Solubility Parameters of 1-Alkyl-3-methylimidazolium Bromide ([C_n_C_1_im]Br, n = 5, 6, 7, 8) Ionic Liquids by Inverse Gas Chromatography and the Hansen Solubility Parameter

**DOI:** 10.3390/molecules24071346

**Published:** 2019-04-05

**Authors:** Qiao-Na Zhu, Qiang Wang, Yan-Biao Hu, Xawkat Abliz

**Affiliations:** 1Center for Physical and Chemical Analysis, Xinjiang University, Urumqi 830046, China; zqnwpc18@163.com (Q.-N.Z.); hnpdshyb@sina.com (Y.-B.H.); 2Key Laboratory of Coal Cleaning Conversion and Chemical Engineering Process, Xinjiang Uyghur Autonomous Region, College of Chemistry and Chemical Engineering, Xinjiang University, Urumqi 830046, China

**Keywords:** ionic liquids, Hansen solubility parameter in practice, Hildebrand solubility parameter, inverse gas chromatography

## Abstract

The physicochemical properties of four 1-alkyl-3-methylimidazolium bromide ([C_n_C_1_im]Br, n = 5, 6, 7, 8) ionic liquids (ILs) were investigated in this work by using inverse gas chromatography (IGC) from 303.15 K to 343.15 K. Twenty-eight organic solvents were used to obtain the physicochemical properties between each IL and solvent via the IGC method, including the specific retention volume and the Flory–Huggins interaction parameter. The Hildebrand solubility parameters of the four [C_n_C_1_im]Br ILs were determined by linear extrapolation to be $\delta_{2([C_{5}C_{1}\mathrm{im}]\mathrm{Br})}$ = 25.78 (J·cm^−3^)^0.5^, $\delta_{2([C_{6}C_{1}\mathrm{im}]\mathrm{Br})}$ = 25.38 (J·cm^−3^)^0.5^, $\delta_{2([C_{7}C_{1}\mathrm{im}]\mathrm{Br})}$ =24.78 (J·cm^−3^)^0.5^ and $\delta_{2([C_{8}C_{1}\mathrm{im}]\mathrm{Br})}$ = 24.23 (J·cm^−3^)^0.5^ at room temperature (298.15 K). At the same time, the Hansen solubility parameters of the four [C_n_C_1_im]Br ILs were simulated by using the Hansen Solubility Parameter in Practice (HSPiP) at room temperature (298.15 K). The results were as follows: $\delta_{t([C_{5}C_{1}\mathrm{im}]\mathrm{Br})}$ = 25.86 (J·cm^−3^)^0.5^, $\delta_{t([C_{6}C_{1}\mathrm{im}]\mathrm{Br})}$ = 25.39 (J·cm^−3^)^0.5^, $\delta_{t([C_{7}C_{1}\mathrm{im}]\mathrm{Br})}$ = 24.81 (J·cm^−3^)^0.5^ and $\delta_{t([C_{8}C_{1}\mathrm{im}]\mathrm{Br})}$ = 24.33 (J·cm^−3^)^0.5^. These values were slightly higher than those obtained by the IGC method, but they only exhibited small errors, covering a range of 0.01 to 0.1 (J·cm^−3^)^0.5^. In addition, the miscibility between the IL and the probe was evaluated by IGC, and it exhibited a basic agreement with the HSPiP. This study confirms that the combination of the two methods can accurately calculate solubility parameters and select solvents.

## 1. Introduction

Ionic liquids (ILs), as demonstrated by green and neoteric solvent research in recent years [1,2], are salts that are commonly made up of asymmetric organic cations, and either inorganic or organic anions [3]. ILs have been widely used as novel electrolytes, separation solvents, and reaction media [4,5], due to their excellent thermal stability, adjustable density, low melting point, strong solvation and high electrochemical stability [6,7]. However, their cost and high viscosity hinders their application. Knowledge about the thermophysical properties of each ionic liquid is essential for scaling up its potential applications. Moreover, 1-alkyl-3-methylimidazolium bromides ([C_n_C_1_im]Br) are one of the most commonly investigated types of ILs, solely for their use as an intermediate compound. Ma [8] separated algae within an entire aquatic ecosystem, according to their stability and high solubility of [C_n_C_1_im]Br (n = 4, 6, 8, 10, 12). In addition, he found that the acute toxicities of these ionic liquids were positively correlated with the alkyl chain length of imidazolium-based ILs. This indicated that [C_12_C_1_im]Br-separated algae was the best, and it could be stored in purified water. Ekka [9] removed Pb(II) from an aqueous solution using [C_n_C_1_im]Br (n = 4, 10, 16) as a template-synthesized mesoporous silica, because [C_n_C_1_im]Br has a high thermal stability, and it is reusable and amphiphilic. The results showed that the ILs bearing a longer alkyl chain [(C_16_C_1_im)Br] were suitable adsorbents for Pb (II) removal, due to their surface area increasing with the increase of carbon chain length. The use of ILs is of great significance in promoting the removal of metals from water.

For a clearer understanding of how the intermolecular interactions of two components influence their applications, many researchers have studied different parameters that are relevant to ILs [10]. The Flory–Huggins interaction parameter ($\chi_{12}$) is a vital tool that is used to predict the thermodynamic state of ILs and select suitable solvents [11]. The Hildebrand solubility parameter ($\delta_{2}$) is a physical and chemical parameter that is inherent to a substance, which is commonly used in the formulation design, chemical additive distribution, solvent selection, and system stability studies and membrane penetration [12,13]. The Hildebrand solubility parameter is usually obtained by dynamic mechanical analysis, titration methods, swelling measurements, group contribution calculation methods, and viscosity measurements [14,15]. However, these methods are often time-consuming and laborious. Thus, inverse gas chromatography (IGC) has been applied to the study of the thermodynamic properties of polymers, carbon blacks, ILs, and other materials [16,17]. In addition, Dr. Charles M. Hansen has proposed the Hansen solubility parameter (HSP) theory, which splits the Hildebrand solubility parameter into three parts: the dispersive interactions, $\delta_{D}$, the polar interactions, $\delta_{P}$, and the hydrogen bonding interactions, $\delta_{H}$. Furthermore, he has developed software known as the Hansen Solubility Parameter in Practice (HSPiP) [18], which uses a genetic algorithm to calculate HSPs [19]. Recently, Ni [20] researched the solubility parameters of alkali lignins using IGC and HSPiP, and found that acetone was a moderately suitable solvent for alkali lignins. It has been confirmed that IGC and HSPiP can be used to determine the solubility parameters of materials, and they are useful for solvent selection. Yu [21] determined the solubility parameters of [C_n_C_1_im][OAC] (n = 2, 4, 6, 8) by IGC and HSPiP, and found that the results were the same. This provides another method for determining the solubility parameters of ILs. Liu [22] has researched the HSP of hydrogenated nitrile rubber (HNBR) by IGC and HSPiP, and has calculated the energy differences (*R*_a_) between HNBR and solvents or solvent mixtures according to their HSP values. In addition, he found that the swelling volume decreases with increasing *R*_a_ values. Therefore, it may be possible to use HSP to predict the swelling phenomena of cured rubber articles in mixed fluids, such as bio-fuels or lubricants. However, solubility parameter data that are related to [C_n_C_1_im]Br (n = 5, 6, 7, 8) have not been reported.

In this study, a widely used method, the IGC method, is proposed for the calculation of the miscibility and $\delta_{2}$ of ILs in various solvents. In addition, HSPiP software is used to calculate the miscibilities and HSPs of four ILs via solubility testing, and a comparative study with the results derived from the IGC is performed.

## 2. Results

### 2.1. Hildebrand Solubility Parameter

#### 2.1.1. Miscibility of the IL and the Probe

The specific retention volume, $V_{g}^{0}$, at the zero-pressure standard state was determined experimentally from Equation (1) [23,24], which is:(1)$$V_{g}^{0}=\frac{273.15}{mT_{a}}F\frac{P_{0}-P_{w}}{P_{0}}(t_{r}-t_{0})\frac{3}{2}\frac{{\left(P_{i}/P_{0}\right)}^{2}-1}{{\left(P_{i}/P_{0}\right)}^{3}-1}$$
where *t*_r_ is the retention time of the probe, *T*_a_ is the room temperature, and *F* is the flow rate of carrier gas, *m* is the mass of the IL on the column, *t*_0_ is the retention time of the non-interacting probe, *P*_w_ represents the saturated vapor pressure, and *P*_0_ and *P*_i_ are the outlet and inlet pressures of the column, respectively.

The specific retention volume, $V_{g}^{0}$, is an important term used in determining the thermodynamic parameters of the IL by the IGC. $V_{g}^{0}$ of 28 probes on four ILs from 303.15 K to 343.15 K were calculated by Equation (1). The results are listed in Appendix A. To obtain the retention graph of the probes, ${\mathrm{In}V}_{g}^{0}$ was plotted with the temperatures from 303.15 K to 343.15 K. For [C_5_C_1_im]Br, Figure 1 demonstrates that the ${\mathrm{In}V}_{g}^{0}$ value decreased with increasing temperature. In addition, a linear relationship between the probe and [C_5_C_1_im]Br was obtained within the range of the experimental temperature. This results indicated that a balance had been established between the probe and [C_5_C_1_im]Br. For the *n*-alkane series, the $V_{g}^{0}$ increased as the numbers of CH_2_ groups increased because of the increase of the interaction forces between the IL and the probe caused by the greater amount of CH_2_ added into the probe.

The Flory–Huggins interaction parameter, $\chi_{12}$, which was obtained using IGC experiments, was calculated by using the expression [13,25].
(2)$$\chi_{12}=\ln(\frac{273.15RV_{2}}{P_{1}^{0}V_{g}^{0}V_{1}})-1-\frac{P_{1}^{0}(B_{11}-V_{1})}{RT}$$
where *V*_2_ is the specific volume of the IL, *T* is the column temperature, *V*_1_ is the molar volume of the probes, $P_{1}^{0}$ represents the probe vapor pressure at the column temperature, *R* represents the gas constant, and *B*_11_ is the second viral coefficient of the probe, where the probe solvent solubility parameter, *δ*_1_, can be obtained from the literature acquired by using Equation (3) [26], which is:(3)$$B_{11}/V_{c}=0.430-0.886(T_{c}/T)-0.694{\left(T_{c}/T\right)}^{2}-0.0375(n-1){\left(T_{c}/T\right)}^{4.5}$$
where *V*_c_ is the critical molar volume of the solvent, *T*_c_ represents the critical temperature of the solvent [27], and *n* is the number of carbon atoms in the solute.

The Flory–Huggins interaction parameter, $\chi_{12}$, plays a significant role in predicting the miscibility between the IL and the probe. The $\chi_{12}$ values were calculated according to Equation (2), as listed in Table 1, which shows that $\chi_{12}$ of some probes, such as thiophene, increased when the temperature increased. However, $\chi_{12}$ of other probes, such as *n*-butyl benzene, *o*-xylene, *m*-xylene, *p*-xylene, ethyl benzene, toluene, nitromethane, methanol, ethanol, *n*-propyl benzene, cyclohexene, octene, pentanone, 3-pentanone, propanol, benzene, isopropanol, butanol, *n*-C_6_ to *n*-C_12_, 2-butanol, and isobutanol decreased when the temperature increased. The reasons for this change in $\chi_{12}$ may include: enthalpy, $\chi_{H}$, and entropic, $\chi_{S}$ [28]. $\chi_{H}$ is related to the intermolecular forces between the IL and the probe, which gradually decrease with increasing temperature. Compared with the enthalpy, $\chi_{S}$ displays an opposite, trend, and it is associated with the free solvent volume. The decrease in the $\chi_{12}$ value means that the IL–probe interactions are becoming strong. According to the Flory–Huggins theory, an $\chi_{12}$ value below 0.5, indicates that the IL and the probe are completely miscible. By contrast, an $\chi_{12}$ value above 0.5 indicates that the IL and probe are insoluble or partially dissolved. In other words, a low $\chi_{12}$ value reflects good compatibility. The following rules have been developed for the system [29,30]: an $\chi_{12}$ value that is lower than 0.5 indicates that the solvent is good, and a value of between 0.5 and 1 indicates a moderately suitable solvent, whereas a $\chi_{12}$ value that is larger than 1 indicates a poor solvent. The results are listed in Table 1. The $\chi_{12}$ values indicated that nitromethane, methanol, ethanol, butanol, thiophene, 2-butanol, isopropanol, propanol, and isobutanol were excellent solvents for all of the examined ILs. By contrast, *n*-propyl benzene, cyclohexene, ethyl benzene, *o*-xylene, *m*-xylene, *p*-xylene, the *n*-alkanes (*n*-C_6_ to *n*-C_12_), octane, and *n*-butyl benzene were poor solvents for all of the examined ILs. Table 1 shows that at the same temperature, the best solvents for dissolving the four ILs were alcohols, followed by benzenes and *n*-alkanes. This finding was related to the polarities of the solvents.

#### 2.1.2. The Hildebrand Solubility Parameter

The Hildebrand solubility parameter is defined as the square root of the cohesive energy (CED) [31].
(4)$$\delta_{1}={\left(\frac{\Delta E_{v}}{V_{1}}\right)}^{1/2}={\left(\frac{\Delta H_{v}-RT}{V_{1}}\right)}^{1/2}={\left(CED\right)}^{1/2}$$
where $\Delta E_{v}$ is the energy of vaporization, *V*_1_ is the molar volume, and $\Delta H_{v}$ is molar enthalpy of evaporation.

For the ILs, the calculation formula for the Hildebrand solubility parameter, *δ*_2_, was calculated using the following equation [32,33]:(5)$$(\frac{\delta_{1}^{2}}{RT}-\frac{\chi_{12}}{V_{1}})=(\frac{2\delta_{2}}{RT})\delta_{1}-\frac{\delta_{2}^{2}}{RT}$$

By plotting the left-hand side of Equation (5) as a function of the probe solubility parameter *δ*_1_ at different temperatures [34], *δ*_2_ was obtained from the slope of the straight line.

The variable *δ*_2_ plays a significant role in selecting solvents to dissolve or swell materials, judging the compatibility of blends, and selecting the pharmaceutical solvents. The *δ*_2_ for the IL [C_5_C_1_im]Br from 303.15 K to 343.15 K was calculated from $\delta_{1}^{2}/RT-\chi_{12}/V_{1}$ versus *δ*_1_, as shown in Figure 2. The $\delta_{1}^{2}/RT-\chi_{12}/V_{1}$ versus *δ*_1_ graphs for the three ILs are shown in Appendix A. The *δ*_2_ of the four examined ILs and the literature are given in Table 2. As shown in Table 2, the increase of temperature from 303.15 to 343.15 K is accompanied by a decrease in the *δ*_2_ of the four ILs, varying within the ranges of 25.71–25.21 (J·cm^−3^)^0.5^, 25.32–24.82 (J·cm^−3^)^0.5^, 24.70–24.22 (J·cm^−3^)^0.5^, and 24.11–23.58 (J·cm^−3^)^0.5^, respectively. The *δ*_2_ shows a slight decrease with increasing temperature, something that has also been observed by Marciniak [35] and Moganty [36]. As the temperature increases, the *δ*_2_ values decrease, because the molar enthalpy of evaporation decreases with temperature, and the molar volume increases with temperature. We also found that *δ*_2_ decreases with increasing alkyl chain length at same temperature, due to the molar enthalpy of evaporation decreasing with the molar mass increase of cations, which is in agreement with the results reported by Alavianmehr [37] and Marciniak [38]. In other words, the more aliphatic the character of the imidazolium cation, the lower the solubility parameters. In addition, we were able to obtain the *δ*_2_ of the ILs at room temperature, using the extrapolation method, based on the relationship curve seen in Figure 3. The *δ*_2_ values of three ILs at 298.15 K are shown in Table 2. They also follow the rule: [C_5_C_1_im]Br > [C_6_C_1_im]Br > [C_7_C_1_im]Br > [C_8_C_1_im]Br.

### 2.2. Hansen Solubility Parameter

According to the HSP concept, the total solubility parameter ($\delta_{t}$) of an IL can be divided into partial solubility parameters, namely, polar ($\delta_{P}$), hydrogen bonding ($\delta_{H}$) and dispersion ($\delta_{D}$) [39,40,41]:(6)$$\delta_{t}^{2}=\delta_{D}^{2}+\delta_{P}^{2}+\delta_{H}^{2}$$

The distance (*R*_a_) between the solvent and the IL within a three-dimensional (3D) diagram was calculated using Equation (7) [42,43]:(7)$$R_{a}={\left[4{\left(\delta_{D}^{1}-\delta_{D}^{2}\right)}^{2}+{\left(\delta_{P}^{1}-\delta_{P}^{2}\right)}^{2}+{\left(\delta_{H}^{1}-\delta_{H}^{2}\right)}^{2}\right]}^{1/2}$$

The relative energy difference (RED), which plays a significant role in predicting the compatibility of the IL and the solvent, can be calculated by Equation (8):(8)$$RED=R_{a}/R_{0}$$
where *R*_0_ is the interaction radius of the IL, *R_a_* is the distance between the solvent and center of the solubility sphere, $\delta_{i}^{1}$ represents the HSP for the IL, and $\delta_{i}^{2}$ is the HSP for the solvent. RED ≤ 1 indicates a good solvent, while a progressively higher RED value implies a poor solvent.

The double-sphere is divided into two domains (the blue solid blue ball at the center represents D_1_, and the green ball represents D_2_). The Hansen solubility parameters of the ILs can be acquired by Equation (9)–(11) [44], which are:(9)$$a=R_{01}/(R_{01}+R_{02})$$
(10)$$b=R_{02}/(R_{01}+R_{02})$$
(11)$$\delta_{i}(\mathrm{Midpoint})=a\times \delta_{i1}+b\times \delta_{i2}$$
where $\delta_{i1}$ and $\delta_{i2}$ are the solubility parameters of the D_1_ and D_2_ domains, *R*_01_ are the interaction radii of the D_1_ domain, *R*_02_ represent the interaction radii of the D_2_ domain, and $\delta_{i}$ (the Midpoint) is a solubility parameter that considers the volume of spheres.

#### 2.2.1. Solubility Test for ILs

The solubility test results of each ionic liquid in 51 pure solvents are summarized in Table 3. We found that good and poor solvents could be obtained from the RED values. The four ILs were found to be poorly dissolved in *n*-propyl benzene, *m*-xylene, cyclohexene, *o*-xylene, ethyl benzene, *p*-xylene, *n*-C_6_, *n*-C_7_, *n*-C_8_, *n*-C_9_, *n*-C_10_, *n*-C_11_, *n*-C_12_, octane, and *n*-butyl benzene, whereas nitromethane, methanol, ethanol, butanol, thiophene, 2-butanol, isopropanol, propanol, and isobutanol were favorable solvents for the four examined ILs. These results are basically consistent with those derived from IGC, based on the $\chi_{12}$ values. The determination of the miscibility between the IL and the probe by HSPiP is a supplement to the determination of IGC, due to the huge amount of HSPiP data. In practical applications, the combination of the two methods can accurately select solvents.

#### 2.2.2. HSPs of the ILs

Considering [C_5_C_1_im]Br as an example, 51 solvents were used to dissolve this. The results of the 3D solubility parameter spheres and the two-dimensional (2D) graphs of the Hansen space are shown in Figure 4. From Figure 4a, we can see that the green sphere is [C_5_C_1_im]Br, the blue solid blue ball at the center represents the D_1_ domain, and the green ball represents D_2_. The blue ball points represent good solvents, and the red data points without the sphere represent solvents, which will, advantageously, not dissolve the IL. Moreover, we can clearly see the solvent distribution from the 2D graphs (Figure 4b). The graphs of HSPs for the other ILs are listed in Appendix A. The simulation results of the four ILs are given in Table 4. The *δ*_t_ values of the four ILs fitted the following rule: [C_5_C_1_im]Br > [C_6_C_1_im]Br > [C_7_C_1_im]Br > [C_8_C_1_im]Br. The *δ*_2_ values of the ILs decreased with an increase of the alkyl chain. It should be added that the values *δ*_2_ of the four ILs obtained by the HSPiP were higher than those obtained by IGC. However, the results obtained by both methods were within an error range of 0.01–0.1 (J·cm^−3^)^0.5^. The harmony between the calculated and experimental values of the solubility parameters is remarkable. The IGC method calculates the *δ*_2_ values through a series of formulas, producing theoretical values, while the HSPiP method is based on solubility testing used to obtain the HSPs of ILs, producing experimental values, which are closer to the true values.

## 3. Materials and Methods

### 3.1. Materials

The 1-pentyl-3-methylimidazolium bromide ([C_5_C_1_im]Br), 1-hexyl-3-methylimidazolium bromide ([C_6_C_1_im]Br), 1-heptyl-3-methylimidazolium bromide ([C_7_C_1_im]Br), and 1-octyl-3-methylimidazolium bromide ([C_8_C_1_im]Br) were supplied by Chengjie Chemical Co., Ltd. (Shanghai, China). The water content and volatile compounds in the ILs were reduced by vacuum evaporation before the experiment. The vacuum evaporation pressure was 0.8 KPa, and the temperature was 363 K. The water content after vacuum evaporation was determined by using the Karl–Fisher titration technique [45], and the mass fraction of the water was less than 600 ppm. The required solvents for the experiment were purchased from J & K Scientific Ltd. (JULABO TitroLine 7750, Germany). All of the studied solvents were used without further purification. The solutes (J & K Scientific Ltd.) with purities better than 0.97 were used without further purification. The CASRN, initial mole fraction purity, initial mole fraction purity, source, and chemical name of the ILs are given in Appendix A.

### 3.2. Inverse Gas Chromatography

All experiments were performed on an Agilent 6890 gas chromatograph (Santa Clara, CA, USA) equipped with a flame ionization detector. The detector temperature was kept at 503.15 K and the injector was operated at 523.15 K during all of the experiments. Methane was used to determine the column holdup time, to calculate the retention times of the various probe solvents. High-purity nitrogen was used as a carrier gas, and the flow rate was 20 mL/min. The oven temperature was varied in 10 K intervals, between 303.15 and 343.15 K. Each experiment was repeated at least three times to check its reproducibility.

The stationary phase used in the experiments was prepared by dissolving a weighed sample of the IL in dichloromethane, and then adding it into a weighed amount of 102 silicon alkylation monomer support (80–100 mesh). The mixture was allowed to dry under a rotary evaporator by slow evaporation, to ensure a homogeneous mixture. The chromatographic column was a stainless steel column, with an inner diameter of 2 mm and a length of 1.2 m, and it was purchased from Dalian Ripley Technological Instruments Co., Ltd. (Dalian, China). The coated support was packed into the stainless steel column, and the stationary phase consisted of 20% IL, which was finally heated for 8 h under nitrogen conditions.

### 3.3. HSPiP Method

To determine the HSP of each IL, its interactions with 51 organic solutes were used to plot the Hansen spheres. A total of 0.2 g IL was placed in a test tube containing 2 mL test solvent. After thorough stirring, the solution was allowed to stand for 24 h at 298.15 K, and dissolution was visually observed. The solvents which could be categorized as good, i.e., those which were totally dissolved in the IL, were given a score of “1”, and poor solvents, which were partially dissolved or insoluble, were given a score of “0”. The experimental data were inputted via HSPiP (Ver.4.1.07, Louisville, KY, USA) to obtain the Hildebrand solubility parameters and HSPs of the ILs.

## 4. Conclusions

It is necessary to know the solubility parameters of ILs. In this study, the *δ*_2_ values of four ILs were calculated by IGC, and the HSPs of the ILs were determined using the HSPiP method, based on solubility testing. It was found that *δ*_2_ decreased with increasing alkyl chain length, as well as when the temperature increased. At room temperature, the *δ*_2_ values of the four [C_n_C_1_im]Br ILs considered were consistent across both methods. In addition, the miscibility between the IL and the probe was successfully determined, using $\chi_{12}$ values and solubility testing, the results were basically consistent across both methods.

## Figures and Tables

**Figure 1 molecules-24-01346-f001:**
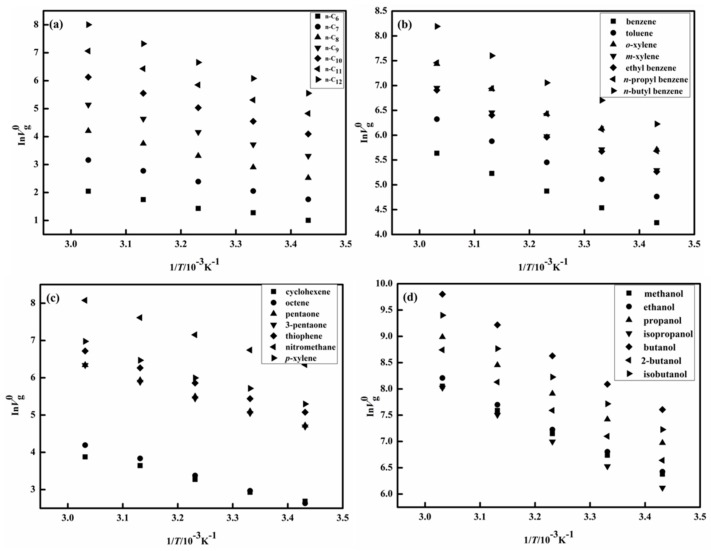
[C_5_C_1_im]Br: Plot of ${\mathrm{In}V}_{g}^{0}$ versus 1000K/T for the probes: (**a**) *n*-C_6_, *n*-C_7_, *n*-C_8_, *n*-C_9_, *n*-C_10_, *n*-C_11_, *n*-C_12_; (**b**) benzene, toluene, *o*-xylene, *m*-xylene, ethyl benzene, *n*-propyl benzene, *n*-butyl benzene; (**c**) cyclohexene, octene, pentanone, 3-pentanone, thiophene, nitromethane, *p*-xylene; (**d**) methanol, ethanol, propanol, isopropanol, butanol, 2-butanol, isobutanol.

**Figure 2 molecules-24-01346-f002:**
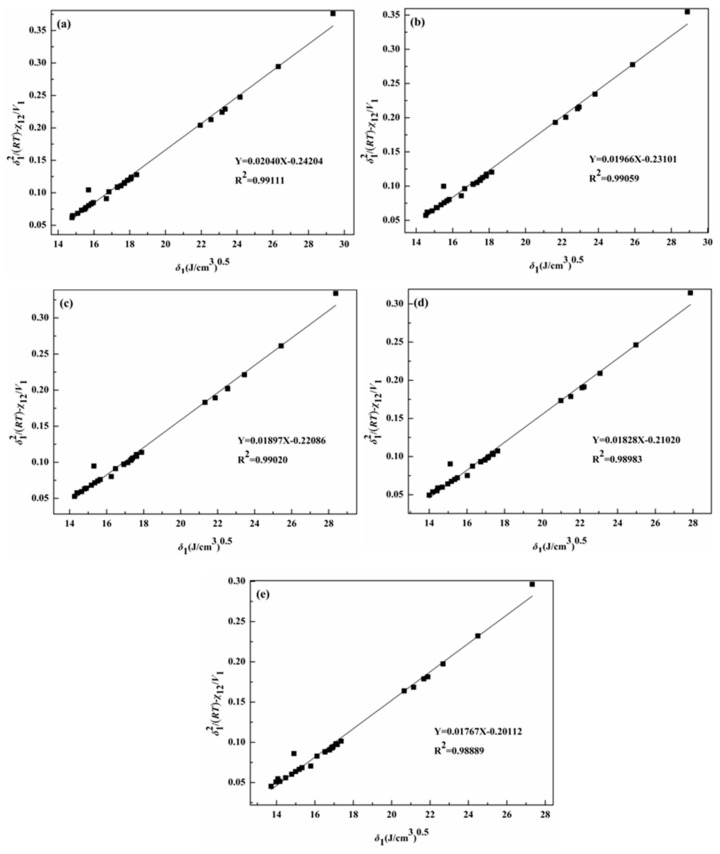
Variation of the term $\delta_{1}^{2}/RT-\chi_{12}/V_{1}$ with solubility parameters of the solvent *δ*_1_ in [C_5_C_1_im]Br (**a**) at 303.15 K; (**b**) at 313.15 K; (**c**) at 323.15 K; (**d**) at 333.15 K; (**e**) at 343.15 K.

**Figure 3 molecules-24-01346-f003:**
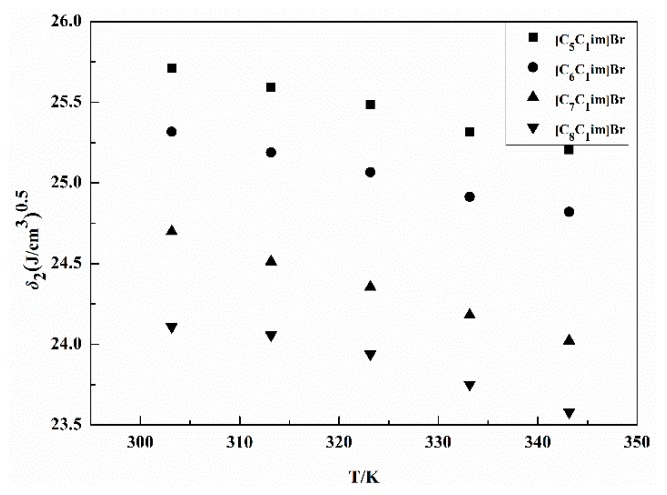
The relation between the solubility parameters of four ILs, *δ*_2_, and the temperatures.

**Figure 4 molecules-24-01346-f004:**
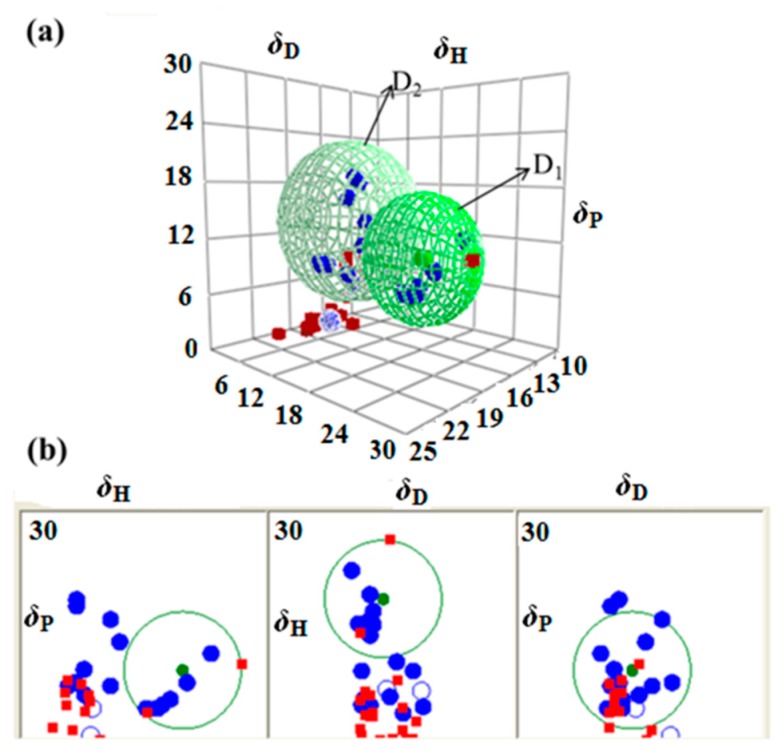
(**a**) The 3D graph with coordinates of [C_5_C_1_im]Br; (**b**)The 2D graphs corresponding to the 3D ones of [C_5_C_1_im]Br.

**Table 1 molecules-24-01346-t001:** The Flory–Huggins interaction parameter, $\chi_{12}$ between the probe and IL at various temperatures for the hypothetical liquid at zero pressure.

Probes	ILs	$\chi_{12}$				
		303.15 K	313.15 K	323.15 K	333.15 K	343.15 K
*n*-C_6_	[C_5_C_1_im]Br	3.33	3.22	3.16	2.96	2.91
	[C_6_C_1_im]Br	3.20	3.16	3.13	2.89	2.80
	[C_7_C_1_im]Br	3.00	2.85	2.82	2.79	2.71
	[C_8_C_1_im]Br	2.38	2.35	2.32	2.29	2.19
*n*-C_7_	[C_5_C_1_im]Br	3.27	3.18	3.13	3.06	2.99
	[C_6_C_1_im]Br	3.19	3.10	3.01	2.97	2.95
	[C_7_C_1_im]Br	2.94	2.81	2.77	2.71	2.65
	[C_8_C_1_im]Br	2.43	2.37	2.33	2.29	2.24
*n*-C_8_	[C_5_C_1_im]Br	3.26	3.19	3.13	3.09	3.04
	[C_6_C_1_im]Br	3.12	3.01	2.94	2.91	2.88
	[C_7_C_1_im]Br	2.97	2.73	2.69	2.67	2.65
	[C_8_C_1_im]Br	2.42	2.39	2.35	2.31	2.28
*n*-C_9_	[C_5_C_1_im]Br	3.39	3.30	3.22	3.16	3.10
	[C_6_C_1_im]Br	3.17	3.06	2.97	2.91	2.83
	[C_7_C_1_im]Br	3.02	2.93	2.84	2.77	2.70
	[C_8_C_1_im]Br	2.61	2.54	2.47	2.40	2.33
*n*-C_10_	[C_5_C_1_im]Br	3.47	3.38	3.28	3.20	3.13
	[C_6_C_1_im]Br	3.26	3.14	3.05	2.98	2.89
	[C_7_C_1_im]Br	3.13	3.02	2.94	2.86	2.79
	[C_8_C_1_im]Br	2.70	2.62	2.54	2.46	2.39
*n*-C_11_	[C_5_C_1_im]Br	3.62	3.51	3.41	3.33	3.23
	[C_6_C_1_im]Br	3.43	3.31	3.22	3.12	3.02
	[C_7_C_1_im]Br	3.31	3.18	3.09	3.00	2.91
	[C_8_C_1_im]Br	2.85	2.76	2.66	2.58	2.50
*n*-C_12_	[C_5_C_1_im]Br	3.76	3.63	3.55	3.44	3.34
	[C_6_C_1_im]Br	3.58	3.47	3.36	3.27	3.17
	[C_7_C_1_im]Br	3.49	3.36	3.26	3.13	3.06
	[C_8_C_1_im]Br	3.04	2.92	2.81	2.70	2.65
benzene	[C_5_C_1_im]Br	0.580	0.551	0.503	0.465	0.414
	[C_6_C_1_im]Br	0.646	0.603	0.582	0.539	0.486
	[C_7_C_1_im]Br	0.695	0.660	0.630	0.607	0.538
	[C_8_C_1_im]Br	0.781	0.758	0.733	0.696	0.640
toluene	[C_5_C_1_im]Br	0.892	0.850	0.822	0.742	0.699
	[C_6_C_1_im]Br	0.973	0.920	0.882	0.783	0.758
	[C_7_C_1_im]Br	1.04	0.980	0.844	0.826	0.808
	[C_8_C_1_im]Br	1.00	0.962	0.947	0.918	0.893
*o*-xylene	[C_5_C_1_im]Br	1.08	1.03	1.01	0.823	0.804
	[C_6_C_1_im]Br	1.14	1.07	1.05	0.909	0.865
	[C_7_C_1_im]Br	1.15	1.10	0.961	0.933	0.918
	[C_8_C_1_im]Br	1.10	1.06	1.03	1.00	0.970
*m*-xylene	[C_5_C_1_im]Br	1.32	1.27	1.24	1.04	1.02
	[C_6_C_1_im]Br	1.34	1.32	1.26	1.11	1.09
	[C_7_C_1_im]Br	1.16	1.11	1.10	1.09	1.07
	[C_8_C_1_im]Br	1.26	1.21	1.18	1.15	1.11
*p*-xylene	[C_5_C_1_im]Br	1.24	1.20	1.17	0.990	0.970
	[C_6_C_1_im]Br	1.27	1.20	1.14	1.06	1.00
	[C_7_C_1_im]Br	1.12	1.08	1.07	1.06	1.04
	[C_8_C_1_im]Br	1.23	1.18	1.16	1.12	1.09
ethyl benzene	[C_5_C_1_im]Br	1.23	1.20	1.13	0.957	0.933
	[C_6_C_1_im]Br	1.26	1.20	1.15	1.04	1.00
	[C_7_C_1_im]Br	1.13	1.05	1.04	1.03	1.00
	[C_8_C_1_im]Br	1.23	1.16	1.13	1.10	1.06
*n*-propyl benzene	[C_5_C_1_im]Br	1.56	1.48	1.44	1.25	1.21
	[C_6_C_1_im]Br	1.26	1.22	1.18	1.16	1.11
	[C_7_C_1_im]Br	1.34	1.29	1.24	1.20	1.17
	[C_8_C_1_im]Br	1.35	1.30	1.26	1.22	1.18
*n*-butyl benzene	[C_5_C_1_im]Br	1.82	1.75	1.69	1.47	1.42
	[C_6_C_1_im]Br	1.46	1.40	1.37	1.33	1.28
	[C_7_C_1_im]Br	1.47	1.41	1.34	1.31	1.28
	[C_8_C_1_im]Br	1.47	1.40	1.36	1.30	1.26
cyclohexene	[C_5_C_1_im]Br	2.28	2.08	2.05	2.02	1.92
	[C_6_C_1_im]Br	2.09	2.01	1.96	1.91	1.87
	[C_7_C_1_im]Br	2.15	2.02	1.98	1.92	1.90
	[C_8_C_1_im]Br	2.13	2.02	1.97	1.92	1.86
octene	[C_5_C_1_im]Br	3.11	2.95	2.93	2.89	2.82
	[C_6_C_1_im]Br	2.86	2.78	2.73	2.68	2.63
	[C_7_C_1_im]Br	2.71	2.59	2.57	2.50	2.47
	[C_8_C_1_im]Br	2.51	2.43	2.40	2.36	2.31
pentanone	[C_5_C_1_im]Br	0.648	0.560	0.536	0.510	0.494
	[C_6_C_1_im]Br	0.805	0.741	0.708	0.666	0.623
	[C_7_C_1_im]Br	0.933	0.842	0.811	0.794	0.775
	[C_8_C_1_im]Br	1.07	1.00	0.977	0.938	0.900
3-pentanone	[C_5_C_1_im]Br	0.836	0.788	0.770	0.739	0.703
	[C_6_C_1_im]Br	0.997	0.939	0.914	0.857	0.820
	[C_7_C_1_im]Br	1.13	1.05	1.01	0.994	0.970
	[C_8_C_1_im]Br	1.27	1.22	1.18	1.14	1.09
thiophene	[C_5_C_1_im]Br	−0.204	−0.196	−0.202	−0.162	−0.153
	[C_6_C_1_im]Br	−0.0210	−0.0170	−0.015	−0.0130	−0.006
	[C_7_C_1_im]Br	0.246	0.252	0.254	0.263	0.264
	[C_8_C_1_im]Br	0.481	0.494	0.538	0.584	0.645
nitromethane	[C_5_C_1_im]Br	−0.358	−0.404	−0.415	−0.443	−0.457
	[C_6_C_1_im]Br	−0.0602	−0.112	−0.140	−0.185	−0.198
	[C_7_C_1_im]Br	0.309	0.243	0.209	0.194	0.167
	[C_8_C_1_im]Br	0.666	0.566	0.503	0.457	0.414
methanol	[C_5_C_1_im]Br	−1.35	−1.39	−1.41	−1.43	−1.47
	[C_6_C_1_im]Br	−1.16	−1.17	−1.21	−1.25	−1.34
	[C_7_C_1_im]Br	−0.762	−0.799	−0.820	−0.841	−0.863
	[C_8_C_1_im]Br	−0.468	−0.543	−0.579	−0.601	−0.636
ethanol	[C_5_C_1_im]Br	−1.13	−1.17	−1.21	−1.26	−1.33
	[C_6_C_1_im]Br	−0.963	−1.01	−1.04	−1.06	−1.07
	[C_7_C_1_im]Br	−0.665	−0.693	−0.740	−0.761	−0.790
	[C_8_C_1_im]Br	−0.401	−0.496	−0.541	−0.575	−0.619
propanol	[C_5_C_1_im]Br	−1.16	−1.25	−1.27	−1.31	−1.35
	[C_6_C_1_im]Br	−1.04	−1.11	−1.17	−1.19	−1.21
	[C_7_C_1_im]Br	−0.815	−0.857	−0.894	−0.921	−0.956
	[C_8_C_1_im]Br	−0.540	−0.664	−0.732	−0.764	−0.816
isopropanol	[C_5_C_1_im]Br	−0.975	−1.032	−1.05	−1.08	−1.13
	[C_6_C_1_im]Br	−0.867	−0.914	−0.955	−0.988	−1.01
	[C_7_C_1_im]Br	−0.627	−0.676	−0.702	−0.722	−0.743
	[C_8_C_1_im]Br	−0.397	−0.487	−0.529	−0.562	−0.599
butanol	[C_5_C_1_im]Br	−1.01	−1.12	−1.17	−1.22	−1.27
	[C_6_C_1_im]Br	−0.889	−0.969	−0.997	−1.03	−1.09
	[C_7_C_1_im]Br	−0.779	−0.839	−0.892	−0.931	−0.999
	[C_8_C_1_im]Br	−0.555	−0.691	−0.752	−0.797	−0.852
2-butanol	[C_5_C_1_im]Br	−1.00	−1.02	−1.06	−1.09	−1.13
	[C_6_C_1_im]Br	−0.863	−0.883	−0.907	−0.955	−1.008
	[C_7_C_1_im]Br	−0.735	−0.781	−0.800	−0.817	−0.834
	[C_8_C_1_im]Br	−0.510	−0.615	−0.658	−0.690	−0.710
isobutanol	[C_5_C_1_im]Br	−1.21	−1.25	−1.33	−1.38	−1.41
	[C_6_C_1_im]Br	−1.05	−1.13	−1.17	−1.20	−1.23
	[C_7_C_1_im]Br	−0.906	−0.982	−1.04	−1.07	−1.10
	[C_8_C_1_im]Br	−0.683	−0.822	−0.892	−0.939	−0.975

Standard uncertainties are as follows: *u* (T) = ± 0.5 K, *u* ($\chi_{12}$) = 0.03.

**Table 2 molecules-24-01346-t002:** The Hildebrand solubility parameter, *δ*_2_, of ILs at various temperatures taken from the literature for the hypothetical liquids at zero pressure.

ILs	T/(K)	*δ*_2_/(J·cm^−3^)^0.5^	Reference
[C_4_C_1_pip][SCN]	298.15	30.70	[35]
	318.15	30.10	
	328.15	29.80	
	338.15	29.50	
	348.15	29.10	
	358.15	28.80	
[C_2_C_1_im][TfO]	283.15	23.10	[36]
	298.15	23.00	
	313.15	22.90	
[C_4_C_1_im][PF_6_]	298.15	29.80	[37]
[C_6_C_1_im][PF_6_]	298.15	28.60	
[C_8_C_1_im][PF_6_]	298.15	27.80	
[C_2_C_1_im][BF_4_]	298.15	24.40	[38]
[C_4_C_1_im][BF_4_]	298.15	24.30	
[C_6_C_1_im][BF_4_]	298.15	23.30	
[C_4_C_1_im][NTf_2_]	298.15	26.70	
[C_6_C_1_im][NTf_2_]	298.15	25.60	
[C_8_C_1_im][NTf_2_]	298.15	25.00	
[C_4_C_1_im][SCN]	298.15	24.64	
[C_6_C_1_im][SCN]	298.15	23.65	
[C_5_C_1_im]Br	298.15	25.78 ^a^	In this work
	303.15	25.71	
	313.15	25.59	
	323.15	25.48	
	333.15	25.32	
	343.15	25.21	
[C_6_C_1_im]Br	298.15	25.38 ^a^	In this work
	303.15	25.32	
	313.15	25.19	
	323.15	25.07	
	333.15	24.91	
	343.15	24.82	
[C_7_C_1_im]Br	298.15	24.78 ^a^	In this work
	303.15	24.70	
	313.15	24.51	
	323.15	24.35	
	333.15	24.18	
	343.15	24.02	
[C_8_C_1_im]Br	298.15	24.23 ^a^	In this work
	303.15	24.11	
	313.15	24.06	
	323.15	23.94	
	333.15	23.75	
	343.15	23.58	

Standard uncertainties are as follows: *u* (T) = ±0.5 K, *u* (*δ*_2_) = 0.02 (J·cm^−3^)^0.5^. ^a^: Obtained by extrapolation

**Table 3 molecules-24-01346-t003:** Solubility test results.

Solvents	HSP/(J·cm^−3^)^0.5^			[C_5_C_1_im]Br		[C_6_C_1_im]Br		[C_7_C_1_im]Br		[C_8_C_1_im]Br	
	*δ* _D_	*δ* _p_	*δ* _H_	Score	RED	Score	RED	Score	RED	Score	RED
acetonitrile	15.3	18.0	6.1	1	0.995	1	0.886	1	0.857	1	1.165 ^a^
2-butanol	15.8	5.7	14.5	1	0.925	1	0.934	1	0.768	1	0.642
butanol	16.0	5.7	15.8	1	0.800	1	0.777	1	0.632	1	0.591
tetrahydrofuran	16.8	5.7	8.0	1	1.043 ^a^	1	0.902	1	0.935	1	0.977
isobutanol	15.1	5.7	15.9	1	0.882	1	0.884	1	0.680	1	0.516
ethylene glycol	17.0	11.0	26.0	1	0.944	1	0.753	1	0.605	1	0.914
isopropanol	15.8	6.1	16.4	1	0.724	1	0.691	1	0.546	1	0.501
dichloromethane	17.0	7.3	7.1	1	0.906	1	0.781	1	0.847	1	0.848
ethanol	15.8	8.8	19.4	1	0.317	1	0.326	1	0.201	1	0.387
pyridine	19.0	8.8	5.9	1	0.799	1	0.751	1	0.856	1	0.756
*N*,*N*-dimethyl formamide	17.4	13.7	11.3	1	0.243	1	0.254	1	0.100	1	0.299
methanol	14.7	12.3	22.3	1	0.778	1	0.718	1	0.527	1	0.794
nitromethane	15.8	18.8	6.1	1	0.967	1	0.882	1	0.850	1	1.152 ^a^
thiophene	18.9	2.4	7.8	1	1.352 ^a^	1	1.361 ^a^	1	1.381 ^a^	1	1.286 ^a^
propanol	16.0	6.8	17.4	1	0.562	1	0.526	1	0.410	1	0.445
acetone	15.5	10.4	7.0	1	0.873	1	0.729	1	0.772	1	0.885
dimethyl sulfoxide	18.4	16.4	10.2	1	0.650	1	0.408	1	0.334	1	0.483
cyclohexanone	17.8	8.4	5.1	1	0.894	1	0.833	1	0.944	1	0.864
propylene oxide	15.2	8.6	6.7	0	1.045	0	1.077	0	1.014	0	1.043
methyl ethyl ketone	16.0	9.0	5.1	0	1.001	0	0.891 ^b^	0	0.966 ^b^	0	1.003
*n*-C_9_	15.7	0.0	0.0	0	2.086	0	2.353	0	2.369	0	2.117
cyclohexene	17.2	1.0	2.0	0	1.787	0	1.831	0	1.865	0	1.786
cyclopentane	16.4	0.0	1.8	0	1.930	0	1.956	0	1.975	0	1.938
3-pentaone	15.8	7.6	4.7	0	1.147	0	1.085	0	1.150	0	1.145
*n*-C_10_	15.7	0.0	0.0	0	2.086	0	2.387	0	2.404	0	2.095
*n*-C_6_	14.9	0.0	0.0	0	2.147	0	2.298	0	2.303	0	2.186
benzene	18.4	0.0	2.0	0	1.862	0	1.896	0	1.925	0	1.859
1,4-dioxane	17.5	1.8	9.0	0	1.404	0	1.416	0	1.414	0	1.339
*n*-C_7_	15.3	0.0	0.0	0	2.115	0	2.306	0	2.317	0	2.149
*o*-xylene	17.8	1.0	3.1	0	1.635	0	2.131	0	1.808	0	1.642
chloroform	17.8	3.1	5.7	0	1.356	0	1.322	0	1.365	0	1.309
*n*-C_12_	16.0	0.0	0.0	0	2.067	0	2.429	0	2.449	0	2.117
*p*-xylene	17.8	1.0	3.1	0	1.705	0	1.805	0	1.841	0	1.690
*n*-butyl benzene	17.4	0.1	1.1	0	1.923	0	2.131	0	2.161	0	1.931
octene	15.3	1.0	2.4	0	1.880	0	2.056	0	2.072	0	1.892
*n*-C_11_	16.0	0.0	0.0	0	2.067	0	2.399	0	2.419	0	2.095
carbon tetrachloride	17.8	0.0	0.6	0	1.956	0	2.010	0	2.039	0	1.968
2,2,4-trimethylpentane	14.1	0.0	0.0	0	2.222	0	2.472	0	2.465	0	2.270
*m*-xylene	17.8	2.6	2.8	0	1.646	0	1.745	0	1.78	0	1.619
ethyl acetate	15.8	5.3	7.2	0	1.207	0	1.167	0	1.194	0	1.169
methyl acetate	15.5	7.2	7.6	0	1.064	0	1.064	0	1.099	0	1.036
methyl propionate	15.5	6.5	7.7	0	1.120	0	1.062	0	1.084	0	1.087
ethyl benzene	17.8	0.6	1.4	0	1.995	0	1.936	0	1.960	0	2.009
toluene	18.0	1.4	2.0	0	1.731	0	1.793	0	1.836	0	1.725
cyclohexane	16.8	0.0	0.2	0	2.012	0	2.095	0	2.120	0	2.033
pentaone	16.0	7.6	4.7	0	1.124	0	1.062	0	1.133	0	1.118
*n*-C_8_	15.5	0.0	0.0	0	2.100	0	2.331	0	2.344	0	2.133
*n*-propyl benzene	17.3	2.2	2.3	0	1.022	0	1.025	0	1.034	0	1.286
methyl formate	15.3	8.4	10.2	0	2.023	0	1.030	0	1.008	0	1.319
*n*-C_5_	14.5	0.0	0.0	0	2.073	0	2.288	0	2.288	0	2.226
ethyl propionate	15.5	6.1	4.9	0	1.283	0	1.261	0	1.308	0	1.275

^a^: Worry out, which means that the Hansen Solubility Parameter in Practice (HSPiP) software prediction should be in the sphere, contrary to the experimental result. ^b^: Worry in, which means that the HSPiP software prediction should be out of the sphere, contrary to the experimental result.

**Table 4 molecules-24-01346-t004:** The HSP and Hildebrand solubility parameters of ILs at room temperature.

ILs	Domain	*δ*/(J·cm^−3^)^0.5^					*R* _0_	Fits
		*δ* _D_	*δ* _P_	*δ* _H_	*δ* _t_	*δ* _2_		
[C_5_C_1_im]Br	D_1_	16.89	10.28	18.87	27.33	-	7.1	0.974
	D_2_	18.40	13.85	10.58	25.34	-	8.7	
	Midpoint	17.72	12.25	14.31	25.86 ^a^	25.78 ^b^	-	
[C_6_C_1_im]Br	D_1_	17.00	8.80	20.30	27.91	-	7.4	0.962
	D_2_	17.90	13.20	9.50	24.18	-	7.9	
	Midpoint	17.46	11.07	14.72	25.39 ^a^	25.38 ^b^	-	
[C_7_C_1_im]Br	D_1_	17.50	7.20	18.60	26.55	-	10.4	0.957
	D_2_	17.50	14.30	8.60	24.16	-	7.9	
	Midpoint	17.50	10.27	14.28	24.81 ^a^	24.75 ^b^	-	
[C_8_C_1_im]Br	D_1_	14.94	9.37	17.12	24.58	-	7.5	0.943
	D_2_	18.43	12.48	10.75	24.72	-	8.2	
	Midpoint	16.76	10.99	13.79	24.33 ^a^	24.23 ^b^	-	

^a^: Obtained by simulation from the double-sphere type. ^b^: Obtained by extrapolation from the IGC data. Standard uncertainties are as follows: *u* (*δ*_t_) = 0.03.

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
