# Peer review of "Practical Determination of the Solubility Parameters of 1-Alkyl-3-methylimidazolium Bromide ([C_n_C_1_im]Br, n = 5, 6, 7, 8) Ionic Liquids by Inverse Gas Chromatography and the Hansen Solubility Parameter"

_molecules, 2019, doi:10.3390/molecules24071346_

Round 1

Reviewer 1 Report

This article describes the use of inverse gas chromatography to determine Flory-Huggins parameters, Hildebrand solubility parameters and Hansen solubility parameters for [CnC1im]Br ionic liquids, where n = 5, 6, 7 and 8, with 28 different solvents. This work is novel in that these parameters have not been enumerated for these ionic liquids and to the best of my knowledge the methodology used to undertake these experiments appear scientifically sound. However, the presentation and interpretation of these results and the presentation of background information in the introduction needs to be substantially improved before this manuscript can be accepted.

Major areas that need to be addressed:

The prior knowledge listed in the introduction simply lists references that have looked at physical properties of [CnC1im]Br with no discussion of the outcomes of their work or the insight they have provided. As a result the precise motivation behind the current work is unclear, such as why the parameters that have been obtained could be potentially useful and how this knowledge could advance the field. The introduction should be rewritten with these factors in mind.

There are similar issues with the results section which largely flow from the lack of clear motivation or aims for the study. It is not clear what the desired outcome of this work is beyond to collect the data. Is it to identify suitable solvents for these ILs, to understand the nature of molecular interactions between these ILs, to work towards developing new separation or extraction systems etc? The meaning behind the data also needs to be made more clear and the results compared to other similar systems in the literature. The exception is the Flory-Huggins section which is fairly clearly written, the Hildebrand and Hansen parameters need to be more clearly described and compared with existing literature results.  

The conclusions will benefit from the changes indicated above as they discuss determining miscibility which is not mentioned elsewhere in the article and not really related to the data obtained. The sentence "According to HSPiP theory, the HSP of ILs can be acquired quickly and accurately" is not really a conclusion. The final statement that these methods provide a powerful theoretical basis for the selection of IL solvents is also not apparent from the preceding work as the discussion focused around the choice of solvent to dissolve ILs and not the other way around. A more consistent chain of logic should be developed.

Minor changes not otherwise discussed:

There are several places where English should be improved as I suspect the intended meaning has changed. Some examples:

Line 33 on page 1 implies high viscosity is a benefit of ILs (it is not).

Line 34 should state "bromides" and "is" on line 35 should be "are"

Line 41 on page 2 "in the solvent mixing influence its applications" should be reworded

Line 43 should state "is a vital tool"

The sentence beginning on line 69 should be reworded as it is unclear.

Line 81 "play" should be "plays"

Line 90 the word "direction," should be removed

The caption of Figure 2 on line 109 is not accurate as the graph depicts the solubility parameters the solvent in [C5C1im]Br and not of [C5C1im]Br itself

A few other minor changes should be made. The abbreviations used should be [CnC1im]Br (numbers subscript) to be consistent with the nomenclature proposed by Hallett and Welton in their 2011 Chemical Reviews paper (DOI: 10.1021/cr1003248) as this is emerging as the predominant nomenclature within the field.

It is stated that "[CnMIM]Br is one of the most commonly investigated ILs". It should be ackowledged here that these are most commonly investigated solely as intermediate and that they are not typically studied as ILs in their own right. This is partially as they are usually solid at room temperature, presumably the ILs used in this study remained supercooled.

When referencing equations that appear in the method section, their location should be referred to as it was not clear where to find Eq(1) which is referenced on page 3 but not presented until page 10. For example ""were calculated by Eq. (1) (materials and methods section). Otherwise, these could be embedded below where they are first introduced which would greatly improve readability.

After these factors have been improved in the text and presentation of the article, I believe there is suitable novelty in the experimental work undertaken to justify its acceptance.  

Author Response

Response to Reviewer Comments

Point 1: The prior knowledge listed in the introduction simply lists references that have looked at physical properties of [CnC1im]Br with no discussion of the outcomes of their work or the insight they have provided. As a result the precise motivation behind the current work is unclear, such as why the parameters that have been obtained could be potentially useful and how this knowledge could advance the field. The introduction should be rewritten with these factors in mind.

Response 1: Thanks for your suggestions. Based on your suggestion, we have revised this introduction.

Ma [1] separated algae within a whole aquatic ecosystem according to their stability and high solubility of [CnC1im]Br (n=4, 6, 8, 10, 12). In addition, he found that the acute toxicity of these ionic liquids was positively correlated with the alkyl chain length of imidazolium-based ILs. This indicated that [C12C1im]Br separated algae which can be stored in purified water is best. Ekka [2] removed Pb(II) from aqueous solution using [CnC1im]Br (n=4, 10, 16) as a template synthesized mesoporous silica, because [CnC1im]Br has thermal stability, reused and amphiphilic. The results showed that the ILs bearing a longer alkyl chain [(C16C1im)Br] was a suitable adsorbent for Pb (II) removal out of all of the adsorbents due to the surface area increasing with increased of carbon chain. The use of ILs is of great significance in promoting the removal of metals from water (see Line 41-50).

Ni [3] researched the solubility parameters of alkali lignin using IGC and HSPiP, and found that acetone was a moderately suitable solvent for the alkali lignin. It has been confirmed that IGC and HSPiP can be used to determine the solubility parameters of materials and are useful for solvent selection. Yu [4] determined the solubility parameters of [CnC1im][OAC] (n=2, 4, 6, 8) by IGC and HSPiP and found that the results were the same. This provides another method for determining the solubility parameters of ILs. Liu [5] has researched the HSP of hydrogenated nitrile rubber (HNBR) by IGC and HSPiP and calculated the energy difference (Ra) between HNBR and solvents or solvent mixtures according to their HSP values. In addition, he found that swelling volume decreases with increasing Ra values. Therefore, it may be possible to use HSP to predict swelling phenomena of cured rubber articles in mixed fluids such as bio-fuels or lubri-cants (see Line 66-76).

In this study, a widely used method, the IGC method is proposed to be able to calculate the miscibility and  of ILs in various solvents. In addition, HSPiP software is used to calculate the miscibility and HSPs of four ILs via solubility testing and a comparative study with the results derived from the IGC is performed (see Line 78-81).

We sent our manuscript to the MDPI to improve the English language. After that, we have revised this manuscript based on the editor’s comment. Here are certificate of English editing.

Point 2: There are similar issues with the results section which largely flow from the lack of clear motivation or aims for the study. It is not clear what the desired outcome of this work is beyond to collect the data. Is it to identify suitable solvents for these ILs, to understand the nature of molecular interactions between these ILs, to work towards developing new separation or extraction systems etc? The meaning behind the data also needs to be made more clear and the results compared to other similar systems in the literature. The exception is the Flory-Huggins section which is fairly clearly written, the Hildebrand and Hansen parameters need to be more clearly described and compared with existing literature results.

Response 2: Thanks for your suggestions. We accept that this suggestion is very helpful in improving the quality of our work. Based on your suggestion, we have revised this introduction.

Based on the introduction, we will divide the discussion into two parts: IGC method is proposed to calculate the miscibility and of ILs in various solvents; the HSPiP software was used to calculate the miscibility and HSPs of four ILs by solubility test and performed a comparative study with the results derived from the IGC (see Results section) .

Point 3: The conclusions will benefit from the changes indicated above as they discuss determining miscibility which is not mentioned elsewhere in the article and not really related to the data obtained. The sentence "According to HSPiP theory, the HSP of ILs can be acquired quickly and accurately" is not really a conclusion. The final statement that these methods provide a powerful theoretical basis for the selection of IL solvents is also not apparent from the preceding work as the discussion focused around the choice of solvent to dissolve ILs and not the other way around. A more consistent chain of logic should be developed.

Response 3: Thanks for your suggestions. Based on your suggestion, we have revised this conclusion.

It is necessary to know the solubility parameter of ILs. In this study, δ2 values of four ILs were calculated by IGC, and the HSPs of the ILs were determined using the HSPiP method based on solubility testing. It was found that δ2 decreased with increasing alkyl chain length, as well as when the temperature increased. At room temperature, the δ2 values of the four [CnC1im]Br ILs considered were consistent across both methods. In addition, the miscibility between the IL and the probe was successfully determined using   values and solubility testing, the results were basically consistent across both methods (see Line 276-283).

Point 4: Line 33 on page 1 implies high viscosity is a benefit of ILs (it is not).

Response 4: Thanks for your suggestions. We find that there are some minor problems in the Line 33 on page 1. High viscosity is not benefit of ILs, so we removed it.

Point 5: Line 34 should state "bromides" and "is" on line 35 should be "are"

Response 5: Thank you for your careful observation and suggestion. Based on your suggestion, we change "is" to "are". We have highlighted text by red color in the revised manuscript

Point 6: Line 41 on page 2 "in the solvent mixing influence its applications" should be reworded

Response 6: Thanks for your suggestions. Based on your suggestion, we have revised this sentence.

of two components influence their application

Point 7: Line 43 should state "is a vital tool"

Response 7: Thanks for your suggestions. We revised "is vital tool" to "is a vital tool" in Line 43.

Point 8: The sentence beginning on line 69 should be reworded as it is unclear.

Response 8: We really appreciate the careful observation of the reviewers. Based on your suggestion, we have revised this sentence.

The specific retention volume,, is an important term used in determining the thermodynamic parameters of the IL by IGC.

Point 9: Line 81 "play" should be "plays"

Response 9: We would like to sincerely thank you for your vigilant observations and critical comments. We really appreciate your valuable suggestion. We revised "play" to "plays" in the Line 81.

Point 10: Line 90 the word "direction," should be removed

Response 10: Thanks for your suggestions. We removed the the word "direction."

Point 11: The caption of Figure 2 on line 109 is not accurate as the graph depicts the solubility parameters the solvent in [C5C1im]Br and not of [C5C1im]Br itself

Response 11: Thanks for your suggestions. Based on your suggestion, we have revised this sentence.

Variation of the term  with solubility parameters of the solvent δ1 in [C5C1im]Br.

Point 12: A few other minor changes should be made. The abbreviations used should be [CnC1im]Br (numbers subscript) to be consistent with the nomenclature proposed by Hallett and Welton in their 2011 Chemical Reviews paper (DOI: 10.1021/cr1003248) as this is emerging as the predominant nomenclature within the field.

Response 12: We would like to sincerely thank you for your vigilant observations and critical comments. We received [CnMIM]Br (n=5, 6, 7, 8) to [CnC1im]Br (n=5, 6, 7, 8).

Point 13: It is stated that "[CnC1im]Br is one of the most commonly investigated ILs". It should be ackowledged here that these are most commonly investigated solely as intermediate and that they are not typically studied as ILs in their own right. This is partially as they are usually solid at room temperature, presumably the ILs used in this study remained supercooled.

Response 13: Thanks for your suggestions. Based on your suggestion, we have revised this sentence.

1-alkyl-3-methylimidazolium bromide ([CnC1im]Br) are one of the most commonly investigated types of IL solely for its use as an intermediate.

Point 14: When referencing equations that appear in the method section, their location should be referred to as it was not clear where to find Eq (1) which is referenced on page 3 but not presented until page 10. For example ""were calculated by Eq. (1) (materials and methods section). Otherwise, these could be embedded below where they are first introduced which would greatly improve readability.

Response 14: Thanks for your suggestions. We moved the Theory section up embedded below where they are first introduced.

1.          Ma, J. M.; Cai, L. L.; Zhang, B. J.; Hu, L. W.; Li, X. Y.; Wang, J. J., Acute toxicity and effects of 1-alkyl-3-methylimidazolium bromide ionic liquids on green algae. Ecotoxicol. Environ. Saf. 2010, 73, (6), 1465-1469.

2.          Ekka, B.; Rout, L.; Kumar, M. K. S. A.; Patel, R. K.; Dash, P., Removal efficiency of Pb(II) from aqueous solution by 1-alkyl-3-methylimidazolium bromide ionic liquid mediated mesoporous silica. J. Environ. Chem. Eng. 2015, 3, (2), 1356-1364.

3.          Ni, H.; Ren, S.; Fang, G.; Ma, Y., Determination of Alkali Lignin Solubility Parameters by Inverse Gas Chromatography and Hansen Solubility Parameters. Bioresources 2016, 11, (2), 4353-4368.

4.          Kaile, Y. U.; Pan, X.; Zhang, Z.; Wang, Q., Determination of Solubility Parameters of Imidazolyl Acetate Ionic Liquid by Inverse Gas Chromatography and Hansen Solubility Parameters. Chemical Gaodeng Xuexiao Huaxue Xuebao 2018, 39, 1048-1054.

5.          Liu, G.; Hoch, M.; Wrana, C.; Kulbaba, K.; Qiu, G., A new way to determine the three-dimensional solubility parameters of hydrogenated nitrile rubber and the predictive power. Polym. Test. 2013, 32, (6), 1128–1134.

Reviewer 2 Report

This manuscript presents a method for determining solvent compatibility in alkyl-imidazolium ILs using IGC. There is a lot of data presented, but no overarching interpretation of the data. There needs to be at least a tentative hypothesis suggested with respect to how δis changing with temperature and with alkyl chain length before this manuscript can be accepted in this journal.

Other, more minor revisions:

- I would recommend moving the Theory section up before the Results and separating it from the experimental section.

- There is no estimation of experimental error provided. I note that the values for δin table 3 are close together - it needs to be clearer that this trend is real within error.

- Figure 4 needs to be rendered in higher resolution.

-"In addition, δof four [CnMIM]Br consists of IGC and HSPiP." this sentence doesn't make sense.

Author Response

Response to Reviewer Comments

Point 1: This manuscript presents a method for determining solvent compatibility in alkyl-imidazolium ILs using IGC. There is a lot of data presented, but no overarching interpretation of the data. There needs to be at least a tentative hypothesis suggested with respect to how δis changing with temperature and with alkyl chain length before this manuscript can be accepted in this journal

Response 1: Thanks for your suggestions. This study was to investigate the Hildebrand solubility parameters and miscibility by IGC technique and HSPiP, therefore we selected 28 different solvents, including alkanes, aromatics, ketones and alcohols to determine the Hildebrand solubility parameter and its fitness R2 is greater than 0.98which indicates that variation of the term  with solubility parameters of the solvent δ1 in IL is valid(see Fig.2 and Figs. S1-S3 in the Supplementary Material). In addition, theoretically the more solvents used in the experiment, the more accurate the HSP data. Therefore, we selected 51 different organic solvents, including alkanes, cycloalkanes, alkenes, ketones, ethers, alcohols, esters and aromatics as well as pyridine, chloroform, nitromethane, dichloromethane, carbon tetrachloride, thiophene, tetrahydrofuran, carbon disulfide , propylene oxide and 1,4-dioxane (see table S2 in the Supplementary Material). And its fitness is greater than 0.94 (see Table 3 in the Manuscript), in other words, the selected solvents are evenly distributed. These solubility parameters values were slightly higher than those obtained by the IGC method, but they only exhibited small errors covering the range of 0.01 to 0.1 (J•cm-3)0.5. In addition, the miscibility between the IL and probe was evaluated by IGC, and it exhibits basically agreement with the HSPiP. This study confirms that the combination of the two methods can accurately calculate solubility parameters and select solvents.

The Hildebrand solubility parameter is defined as the square root of the cohesive energy (CED) [1]

where is the energy of vaporization, V1 is the molar volume andis molar enthalpy of evaporation.

As shown in Table 2 (see Line 174-177), the increase of temperature from 303.15 to 343.15 K is accompanied by a decrease in the δ2 of the four ILs, varying within the ranges 25.71-25.21 (J•cm-3)0.5, 25.32-24.82 (J•cm-3)0.5, 24.70-24.22 (J•cm-3)0.5 and 24.11-23.58 (J•cm-3)0.5, respectively. The δ2 shows a slight decrease with increasing temperature, something which has also been observed by Marciniak [2] and Moganty [3]. As the temperature increases, the δ2 values decrease because the molar enthalpy of evaporation decreases with temperature and the molar volume increases with temperature. We also found that δ2 decreases with increasing alkyl chain length at same temperature due to the molar enthalpy of evaporation decreasing with the molar mass increase of cations, which is in agreement with the results reported by Alavianmehr [4] and Marciniak [5]. In other words, the more aliphatic the character of the imidazolium cation leads to the lower the solubility parameters.

 We sent our manuscript to the MDPI to improve the English language. After that, we have revised this manuscript based on the editor’s comment. Here are certificate of English editing.

Point 2: I would recommend moving the Theory section up before the Results and separating it from the experimental section.

Response 2: Thanks for your suggestions. We moved the Theory section up embedded below where they are first introduced.

Point 3: There is no estimation of experimental error provided. I note that the values for δin table 3 are close together - it needs to be clearer that this trend is real within error.

Response 3: Thanks for your suggestions. The experimental standard uncertainty is 0.03.

Point 4: Figure 4 needs to be rendered in higher resolution.

Response 4: Thanks for your suggestions. We improved the resolution of Figure 4 (see Figure 4).

Point 5: "In addition, δ2 of four [CnMIM]Br consists of IGC and HSPiP." this sentence doesn't make sense.

Response 5: Thanks for your suggestions. Based on your suggestion, we have revised this conclusion.

It is necessary to know the solubility parameter of ILs. In this study, δ2 values of four ILs were calculated by IGC, and the HSPs of the ILs were determined using the HSPiP method based on solubility testing. It was found that δ2 decreased with increasing alkyl chain length, as well as when the temperature increased. At room temperature, the δ2 values of the four [CnC1im]Br ILs considered were consistent across both methods. In addition, the miscibility between the IL and the probe was successfully determined using   values and solubility testing, the results were basically consistent across both methods (see Line 276-283).

1.      Weerachanchai, P.; Chen, Z.; Leong, S. S. J.; Chang, M. W.; Lee, J. M., Hildebrand solubility parameters of ionic liquids: Effects of ionic liquid type, temperature and DMA fraction in ionic liquid. Chem. Eng. J. 2012, 213, (12), 356-362.

2.      Marciniak, A., The Hildebrand Solubility Parameters of Ionic Liquids—Part 2. Int. J. Mol. Sci. 2011, 12, (12), 3553-3575.

3.      Moganty, S. S.; Baltus, R. E., Regular Solution Theory for Low Pressure Carbon Dioxide Solubility in Room Temperature Ionic Liquids: Ionic Liquid Solubility Parameter from Activation Energy of Viscosity. Ind. Eng. Chem. Res. 2010, 49, (12), 5846-5853.

4.      Alavianmehr, M. M.; Hosseini, S. M.; Mohsenipour, A. A.; Moghadasi, J., Further property of ionic liquids: Hildebrand solubility parameter from new molecular thermodynamic model. J. Mol. Liq. 2016, 218, 332-341.

5.      Marciniak, A., The solubility parameters of ionic liquids. Int. J. Mol. Sci. 2010, 11, (5), 1973-1990.

Round 2

Reviewer 2 Report

The manuscript is much improved after restructuring and is now suitable for publication.